# DNA Damage Regulates the Functions of the RNA Binding Protein Sam68 through ATM-Dependent Phosphorylation

**DOI:** 10.3390/cancers14163847

**Published:** 2022-08-09

**Authors:** Venturina Stagni, Silvia Orecchia, Luca Mignini, Sara Beji, Ambra Antonioni, Cinzia Caggiano, Daniela Barilà, Pamela Bielli, Claudio Sette

**Affiliations:** 1Institute of Molecular Biology and Pathology, National Research Council (CNR), 00185 Rome, Italy; 2IRCCS Fondazione Santa Lucia, 00143 Rome, Italy; 3Department of Biomedicine and Prevention, University of Rome Tor Vergata, 00133 Rome, Italy; 4Department of Neuroscience, Section of Human Anatomy, Catholic University of the Sacred Heart, 00168 Rome, Italy; 5G-STeP-Organoids Research Core Facility, Fondazione Policlinico Universitario A. Gemelli, IRCCS, 00168 Rome, Italy; 6Department of Biology, University of Rome Tor Vergata, 00133 Rome, Italy

**Keywords:** DNA damage, ATM kinase, Sam68, alternative splicing, alternative polyadenylation

## Abstract

**Simple Summary:**

Alterations of the complex network of interactions between the DNA damage response pathway and RNA metabolism have been described in several tumors, and increasing efforts are devoted to the elucidation of the molecular mechanisms involved in this network. Previous large-scale proteomic studies identified the RNA binding protein Sam68 as a putative target of the ATM kinase. Herein, we demonstrate that ATM phosphorylates Sam68 upon DNA damage induction, and this post-translational modification regulates both the signaling function of Sam68 in the initial phase of the DNA damage response and its RNA processing activity. Thus, our study uncovers anew crosstalk between ATM and Sam68, which may represent a paradigm for the functional interaction between the DDR pathway and RNA binding proteins, and a possible actionabletarget in human cancers.

**Abstract:**

Cancer cells frequently exhibit dysregulation of the DNA damage response (DDR), genomic instability, and altered RNA metabolism. Recent genome-wide studies have strongly suggested an interaction between the pathways involved in the cellular response to DDR and in the regulation of RNA metabolism, but the molecular mechanism(s) involved in this crosstalk are largely unknown. Herein, we found that activation of the DDR kinase ATM promotes its interaction with Sam68, leading to phosphorylation of this multifunctional RNA binding protein (RBP) on three residues: threonine 61, serine 388 and serine 390. Moreover, we demonstrate that ATM-dependent phosphorylation of threonine 61 promotes the function of Sam68 in the DDR pathway and enhances its RNA processing activity. Importantly, ATM-mediated phosphorylation of Sam68 in prostate cancer cells modulates alternative polyadenylation of transcripts that are targets of Sam68, supporting the notion that the ATM–Sam68 axis exerts a multifaceted role in the response to DNA damage. Thus, our work validates Sam68 as an ATM kinase substrate and uncovers an unexpected bidirectional interplay between ATM and Sam68, which couples the DDR pathway to modulation of RNA metabolism in response to genotoxic stress.

## 1. Introduction

The human genome is exposed to a multitude of DNA damaging agents that can severely compromise its integrity. If not properly repaired, accumulation of DNA lesions in the cell upon ageing may affect a variety of cellular processes and can lead to pathological conditions, such as neurodegenerative diseases and carcinogenesis [1,2,3]. Genome integrity is maintained by several tightly controlled surveillance mechanisms that are collectively termed the DNA damage response (DDR) system. Recognition of single- and double-strand DNA lesions recruits and activates, respectively, two DDR kinases: ataxia–telangiectasia- and RAD3-related (ATR) and ataxia–telangiectasia mutated (ATM). Once activated, these kinases phosphorylate downstream substrates in the pathway. Among others, the serine/threonine kinases CHK1 and CHK2 play a crucial role, by transmitting the signal to checkpoint effectors that coordinate DNA repair with cellular processes [4].

Mounting evidence points to the existence of a tight interplay between the DDR pathway and RNA metabolism [5,6,7]. DNA damage influences RNA metabolism, particularly transcription and splicing, as well as the activity and intracellular localization of RNA binding proteins (RBPs) [8,9,10,11,12,13,14,15]. Moreover, factors involved in DNA repair can also sustain the DDR by regulating RNA processing events [14,16,17,18,19,20]. For instance, in response to DNA damage, the BRCA1 protein forms a complex with BCLAF1 and other splicing factors to regulate splicing of DDR genes, thus promoting DNA repair and genome stability [16]. On the other hand, several RBPs and RNA processing factors have been reported to play an active role in the DDR pathway [6]. CDK12, a cyclin-dependent kinase that phosphorylates the RNA polymerase II [21], ensures proper expression of DDR genes by suppressing intronic alternative polyadenylation and premature transcript termination [22,23], whereas the inositol polyphosphate multikinase (IPMK) guarantees nuclear export of transcripts involved in DNA repair [24]. RBPs can also take part in the DDR, as exemplified by hnRNPD, which binds chromatin and re-localizes to DNA breaks where it promotes DDR signaling and DNA end-resection [25]. These observations imply the existence of a coordinated interplay between DDR proteins and RNA processing factors, which contributes to maintenance of genome integrity through both enhancement of the DNA repair process and induction of genes involved in this pathway. Nevertheless, the mechanisms underlying this interplay are still largely unknown.

The Src-associated in mitosis of 68 KDa (Sam68) protein, also known as KHDRBS1, is a multifunctional RBP belonging to thesignal transduction and activation of RNA (STAR) family [26,27]. Sam68 acts as scaffold protein to participate in the regulation of signal transduction pathways induced by extracellular signals [27,28]. Moreover, Sam68 modulates several steps in gene expression regulation through its RNA binding activity [26,27]. The pleiotropic functions of Sam68 imply that its expression and activity must be finely regulated under physiological conditions [29,30,31,32,33]. On the other hand, aberrant Sam68 expression has been correlated with malignant features and poor prognosis in a variety of human cancers [26,34,35,36,37,38,39]. The pro-oncogenic effects of Sam68 were associated withall its multiple functions, from intracellular signaling [34] to transcriptional [35,36,37,38] and splicing regulation [39,40,41]. More recently, a link between Sam68 and the DDR pathway was also uncovered. Sam68 is recruited to DNA lesions, where it interacts with PARP1 and promotes PAR production. Depletion of Sam68 caused hypersensitivity to genotoxic stress [42], indicating its crucial role in the DDR pathway. Moreover, Sam68 was shown to be necessary for PAR-dependent activation of NFκB upon colon cell irradiation, a pathway that conferred radioprotection to colon epithelium and colorectal cancer cells in vivo [43,44].

Sam68 plays a particularly important role in prostate cancer (PC). Sam68 is up-regulated in PC cells and patients [45], interacts with the androgen receptor [35], and promotes splicing of oncogenic variants of cancer-associated genes [39,46,47]. Although Sam68 was previously shown to promote chemoresistance in PC cells [45], its specific role in the DDR pathway was not investigated. Herein, we identified Sam68 as a novel substrate of the ATM kinase in PC cells. ATM interacts with Sam68 and phosphorylates it in serine/threonine (S/T) residues, thus selectively modulating its ability to bind PARP1 and RNA. Functionally, we provide evidence that ATM-mediated phosphorylation of Sam68 is required for DDR and NFκB activation and to regulate alternative polyadenylation of target genes in PC cells. Thus, our study uncovers an unexpected coordination of Sam68 functions by ATM and highlights a new link between DDR transduction and gene expression regulation in PC cells.

## 2. Materials and Methods

### 2.1. DNA Constructs

GFP N-terminal tagged Sam68-WT and Flag C-terminal tagged ATM plasmids have been already described [48,49]. The pCDNA GFP SAM68 T61A, S388A, and S390A constructs were generated using the QuickChange site-directed mutagenesis kit (Stratagene, San Diego, CA, USA) using pCDNA-GFP-SAM68-wt as template.

### 2.2. Antibodies and Other Reagents

The following antibodies and reagents were used: anti–phospho-Ser1981-ATM, anti-phospho-Chk2(T68), anti-Chk2, anti-phopsho-S15-p53, anti-PARP1 (all from Cell Signaling, Beverly, MA, USA); anti-ATM (2C1), anti-p53 (DO-1), anti-HSP90 (F8), anti-actin B, anti-vinculin, anti-GAPDH (all from Santa Cruz Biotechnology, Santa Cruz, CA, USA); anti-PAR (Ab-1; Sigma Aldrich, St. Louis, MO, USA, cat no AM80); anti-phospho-S824 Kap1 (Abcam, Cambridge, UK, ab70369); anti-SAM68 (purified); neocarzinostatin (NCS) (Sigma-Aldrich, St. Louis, MO, USA, cat no N9162), a radiomimetic chemotherapy drug, was used at 500 ng/mL concentration, and KU55933 (Sigma Aldrich, St. Louis, MO, USA, cat no SML11109), a specific inhibitor of ATM kinase, was used at 10 μM concentration.

### 2.3. Cell Culture and Transfections

Human embryonic kidney cell line HEK293T was grown in sterile culture dishes in Dulbecco’s Modified Eagle Medium (DMEM) (Sigma-Aldrich, St. Louis, MO, USA) supplemented with 10% fetal bovine serum (Sigma-Aldrich, St. Louis, MO, USA), streptomycin 10 mg/mL, penicillin 10,000 U/mL, and L-glutamine 1% (Sigma-Aldrich, St. Louis, MO, USA) at 37 °C and 5% CO_2_. HEK-293T cells were transfected with Lipofectamine 2000 (Invitrogen, Waltham, MA, USA) for 24 h before protein extraction. Human prostatic carcinoma cell lines LNCaP and 22Rv1 were grown in RPMI 10% fetal bovine serum (Sigma-Aldrich, St. Louis, MO, USA), streptomycin 10 mg/mL, penicillin 10,000 U/mL, and L-glutamine 1% (Sigma-Aldrich, St. Louis, MO, USA) at 37 °C and 5% CO_2_. RNA interference cells were infected with lentiviral particles produced using Mission^®^ shRNA SAM68, TRCN0000000048, Clone ID: NM_006559.x-527s1c1 plasmid (Sigma-Aldrich, St. Louis, MO, USA).

### 2.4. Immunoblotting and Immunoprecipitation

Cell extracts were prepared in lysis buffer (50 mM Tris-HCl, pH7.5 250 mM NaCl, 1% NP-40, 5 mM EDTA, 5 mM EGTA, 1 mM phenylmethylsulfonyl fluoride, 25 mM NaF, 1 mM orthovanadate, 10 mg/mL TPCK, 5 mg/mL TLCK, 1 mg/mL leupeptin, 10 mg/mL soybeantrypsininhibitor,1 mg/mL aprotinin). For immunoblotting, protein extracts (50 µg) were separated by SDS–polyacrylamide gel electrophoresis, blotted onto nitrocellulose membrane, and detected with specific antibodies. For immunoprecipitation, protein extracts prepared in lysis buffer were pre-cleared to reduce non-specific binding of proteins to Sepharose beads. The samples were incubated with specific antibodies against Flag-tag, GFP-tag, endogenous Sam68 and ATM; after 2 h, G-or A-coupled Sepharose beads were added to precipitate the antibody/antigen complex (Amersham Biosciences Co., Buckinghamshire, UK). Immunocomplex wasresolved and analyzed by SDS–polyacrylamide gel electrophoresis. All immunoblots were revealed by enhanced chemiluminescence (Amersham Biosciences Co., Buckinghamshire, UK).Original Western blots used for the figures are shown in the Appendix A.

### 2.5. PolyA Pulldown Assay

PolyA pulldown assay was performed as previously described [46]. Briefly, 293T cells were incubated in lysis buffer [50 mM Hepes, pH 7.4, 150 mM NaCl, 15 mM MgCl_2_, 10% glycerol, 1 mM dithiothreitol, 20 mM β-glycerophosphate, 0.5 mM NaVO_4_, protease inhibitor cocktail (Sigma-Aldrich, St. Louis, MO, USA), 0.5% Triton X-100] for 10 min in ice. Cell suspension was centrifuged for 10 min at 12,000× *g* at 4 °C, and the supernatant fraction was collected (cell extract). Cell extracts (300 µg each) were precleared for 1 h on streptavidin–Sepharose beads (Sigma-Aldrich, St. Louis, MO, USA), while PolyA Sepharose 4B beads (Amersham Biosciences Co., Buckinghamshire, UK) were incubated in lysis buffer supplemented with 1% BSA for 1 h and then washed. Pre-cleared extracts were then incubated with BSA-blocked PolyA Sepharose 4B. After 2 h of incubation at 4 °C under rotation, beads were washed three times with lysis buffer and proteins were eluted in SDS sample buffer for western blot analysis.

### 2.6. Survival Assay

For survival assays, HEK293T cells were transfected with Lipofectamine 2000 (Invitrogen, Waltham, MA, USA) for 24 h and then treated with NCS 500 µg/mL. Viability was assessed 48 h after treatment by the Cell Titer GloLuminescent Assay (Promega, Madison, WI, USA).

### 2.7. RT-PCR and qPCR Analyses

Total RNA was extracted using TRIzol reagent (Invitrogen, Waltham, MA, USA) according to the manufacturer’s instructions. After digestion with RNase-free DNase (Invitrogen, Waltham, MA, USA), RNA was resuspended in RNase-free water (Sigma-Aldrich, St. Louis, MO, USA) and retrotranscribed (1 µg) using M-MLV reverse transcriptase (Promega, Madison, Wisconsin, USA). As template, 40 nanograms of cDNA was used for both conventional PCR (RT-PCR) (GoTaq, Promega, Madison, WI, USA) and quantitative real-time PCR (qPCR) (SYBR Green, Roche, Basel, Switzerland) analyses. Primers used for PCR reactions are listed in Appendix A.

### 2.8. Immunofluorescence Analysis

Cells were washed, seeded on glass cover slips by centrifugation, and fixed in 3.7% formaldehyde for 10 min at room temperature. Cells were permeabilized in 0.25% Triton X-100 for 5 min at room temperature and 100% methanol for 10 min at −20 °C and pre-incubated in 20% goat serum for 30 min at 37 °C in humidified chamber. Next, cells were incubated for 2 h with mouse anti-phospho-Ser1981- ATM (1:200, Cell Signaling, Beverly, MA, USA). After three washes in PBS, cells were incubated with rhodamine-conjugated anti-rabbit antibody (1:600 dilution, sc2091; Santa Cruz Biotechnology, Santa Cruz, CA, USA). Incubation with 0.1 µg/mL 4′,6-diamidino-2-phenylindole (DAPI) for 1 min at room temperature was performed to stain DNA. Samples were examined under an AX70 microscope (Olympus, Tokyo, Japan) equipped with epifluorescence, and photographs were taken with a charge-coupled device camera (Photometrics, Tucson, AZ, USA). ImageJ software was used to count foci and overall fluorescence intensity.

### 2.9. Quantification and Statistical Analysis

Densitometric analyses of western blot films and agarose gels were performed using ImageJ software. Statistical significance was calculated by Student’s *t*-test on at least three independent experiments; *p*-values are represented as follows: *, *p* ≤ 0.05; **, *p* < 0.01; ***, *p* < 0.001; n.s., *p* > 0.05.

## 3. Results

### 3.1. Sam68 Is a Substrate of the ATM Kinase in Prostate Cancer Cells

Sam68 was previously reported to promote the DDR signaling pathway in some cellular systems [42,43,44]. To test whether Sam68 is also involved in activation of the DDR pathway in PC cells, we depleted its expression in androgen-deprivation-sensitive (LNCaP) and -resistant (22Rv1) cells by lentiviral transduction of a short hairpin RNA (shRNA). PC cells were then treated with the radiomimetic drug neocarzinostatin (NCS) to induce DNA damage. First, we assessed that silencing of Sam68 was maintained for the whole time-course (Appendix A). Next, we evaluated the phosphorylation/activation of ATM as a marker of DDR induction. Western blot analyses indicated that phosphorylation of ATM was reduced in LNCaP and 22Rv1 cells depleted of Sam68, whereas no changes in the total expression levels of ATM wereobserved (Appendix A). The inhibition of ATM phosphorylation was more pronounced in 22Rv1 cells than in LNCaP cells, in line with higher efficiency of Sam68 depletion in the former cell line (Appendix A). In response to DDR activation, phosphorylated ATM is recruited to foci of DNA lesions to induce the repair pathway [1,2]. We found that NCS induced the formation of foci of phosphorylated (pS1981) ATM in the majority of LNCaP cells, an effect that was repressed by the PARP inhibitor olaparib (Figure 1A,C,D). An even stronger reduction in pS1981-ATM foci and overall nuclear fluorescence intensity was observed in Sam68-depleted LNCaP cells treated with NCS (Figure 1A–D). Thus, as previously observed in other cell types [42,43,44], Sam68 is required for proper activation of the DDR pathway in PC cells as well.

A phosphoproteomic screen identified Sam68 as one of the putative substrates of ATM in response to DNA damage [48]. Nevertheless, the possible functional interaction between the two proteins was not validated. To test whether Sam68 and ATM physically interact, we transiently transfected HEK293T cells with constructs that overexpress GFP-Sam68 and Flag-ATM. Western blot analysis of whole-cell lysates (WCL) indicated that overexpression of ATM leads to ATM activation, as shown by increased phosphorylation of the kinase (Figure 2A, left panel; pS1981-ATM signal, compare lane 1 and 3). Moreover, reciprocal co-immunoprecipitation assays using these WCL samples indicated that recombinant ATM and Sam68 proteins physically interact (Figure 2A, right panels). Physical interaction was also observed between the endogenous ATM and Sam68 proteins in LNCaP cells (Figure 2B). Interestingly, this interaction was markedly increased upon treatment of cells with NCS, which leads to ATM autophosphorylation and activation (WCL panels; Figure 2B), whereas it was completely suppressed upon inhibition of ATM kinase activity by the specific inhibitor KU55933 (IP panels; Figure 2B).

Next, to test if Sam68 is a substrate of ATM, GFP-Sam68 phosphorylation was evaluated using a pSQ/TQ antibody. This antibody detects phosphorylated serine (S) or threonine (T) residues followed by a glutamine (Q), which represents the consensus motif for ATM kinase, as well as ATR and DNA-PK [49]. We noticed that overexpression of GFP-Sam68 resulted in activation of the endogenous ATM (Figure 2C; WCL pS1981 panel, compare lanes 1 and 4). Moreover, immunoprecipitation experiments revealed that GFP-Sam68 was efficiently phosphorylated on SQ/TQ residues and that this effect relied on ATM activity, as it was suppressed by treatments with KU55933 (Figure 2C, right panels). Immunoprecipitation of the endogenous protein from LNCaP cells indicated that Sam68 is phosphorylated on pSQ/TQ residues also in PC cells and that this post-translational modification is increased by NCS treatment (Figure 2D). Moreover, Sam68 phosphorylation depends on the kinase activity of ATM, as it was inhibited by treatment with KU55933 (Figure 2D). These results identify Sam68 as a new ATM-interacting protein which is phosphorylated by the kinase.

### 3.2. Threonine 61 in Sam68 Is a Specific Target of ATM Kinase Activity

Sam68 is characterized by several structural and regulatory motifs (Figure 3A) [50]. The GLD-1/Sam68/GRP33 (GSG) homology domain in the central part of the protein includes the heterogeneous nuclear ribonucleoprotein (hnRNP) K homology (KH) motif that is required for RNA binding and homo-dimerization [50]. The regions flanking the GSG domain comprise a phosphotyrosine-rich domain (YY) and six proline-rich sequences (P0-P5) that allow interaction with proteins containing, respectively, Src Homology 2 (SH2) and 3 (SH3) domains [50,51], while its nuclear localization signal is embedded in the C-terminal region [52]. Inspection of the human Sam68 protein sequence revealed three S/T–Q motifs (T61, S388, S390) that may represent potential ATM target sites (Figure 3A). These motifs exhibit 100% identity in the sequence of mouse and rat Sam68, suggesting their functional relevance. To test whether these residues are targeted by ATM-dependent phosphorylation, we generated non-phosphorylatable mutants by substituting S and T residues with alanine (A). First, we focused on T61, as this residue was previously identified in the screen for substrates phosphorylated by ATM upon DNA damage induction [48]. Immunoprecipitation experiments using the anti-pSQ/TQ antibody and extracts from HEK293T cells treated with NCS confirmed the phosphorylation of wild-type Sam68 (GFP-Sam68_wt_). By contrast, immunoprecipitation of the GFP-Sam68_T61A_ was reduced with respect to the wild-type protein (Figure 3B, IP SQ/TQ panels), even though this protein was expressed at similar levels as GFP-Sam68_wt_ and the ATM pathway (p1981-ATM and pS15p53) was induced in response to NCS (Figure 3B, WCL panels). This effect was specific as no differences in immunoprecipitation of the known ATM substrate p53 were observed in all NCS-treated samples (Figure 3B). A similar reduction in GFP-Sam68_T61A_ phosphorylation upon NCS treatment was also observed when GFP-Sam68 protein was immunoprecipitated and analyzed by anti-pSQ/TQ western blot (Figure 3C, compare lanes 2 and 3). This approach indicated that the SQ motifs downstream of the GSG are also substrates of ATM activity, as shown by the reduced phosphorylation of the GFP-Sam68_S388A/S390A_ and GFP-Sam68_T61A/S388A/S390A_ mutants in cells treated with NCS (Figure 3C). These findings indicate that Sam68 is a novel substrate of ATM and identify three S/T residues phosphorylated by the kinase in response to DNA damage.

### 3.3. ATM Promotes the Interaction of Sam68 with PARP1 through Threonine 61 Phosphorylation

We then sought to investigate the functional link between ATM and Sam68. Sam68 interacts with PARP1 and is recruited to DNA lesions. Moreover, the Sam68/PARP1 interaction is crucial for DNA damage-initiated, PARP1-dependent PARylation [42]. T61 resides in the N-terminal region of Sam68 (Figure 3A), which is required for the interaction with PARP1 and the downstream activation of NFκB upon DDR induction [42,53]. Thus, we asked if ATM-dependent phosphorylation of T61 could selectively regulate the Sam68–PARP1–NFκB axis. To this end, HEK293T cells were transiently transfected with GFP-Sam68_wt_ or GFP-Sam68_T61A_ and treated with NCS for different time points. DDR induction by NCS was positively correlated with an early and transient interaction of Sam68 with both ATM and PARP1 (Figure 4A), and with an increase in total PAR chain formation in WCL (Figure 4B). Notably, both these events were reduced in the presence of the GFP-Sam68_T61A_ mutant, indicating that phosphorylation of T61 is required for the functional interaction of Sam68 and PARP1 in response to DNA damage. Accordingly, we observed that overexpression of the GFP-Sam68_T61A_ mutant impaired signaling downstream of ATM, as NCS-induced phosphorylation of NFκB was reduced and less sustained over time (Figure 4B). A similar effect was observed for pS824 KAP1 (Figure 4C; WCL panels), p53 and, albeit to a lesser extent, CHK2 phosphorylation (Figure 4B). Notably, the GFP-Sam68_S388A/S390A_ mutant was still capable of interacting with PARP1 (Figure 4C; IP:GFP panels) and inducing its activity in response to DNA damage, leading to NFκB and KAP1 phosphorylation comparable to that observed in the presence of GFP-Sam68_wt_ (Figure 4C; WCL panels). However, protein PARylation was partially reduced in cells expressing the GFP-Sam68_S388A/S390A_ mutant (Figure 4C; WCL panels). These results indicate that the functional interaction between Sam68 and PARP1 is primarily modulated by ATM-dependent phosphorylation of T61. In line with this notion, expression of GFP-Sam68_wt_, and to a lesser extent of the GFP-Sam68_S388A/S390A_ mutant, rescued viability of cells treated with NCS, whereas cells expressing GFP-Sam68_T61A_ displayed reduced viability even when compared to GFP-expressing control cells (Figure 4D). These data provide compelling evidence that impairing ATM-dependent phosphorylation of Sam68 on T61 disrupts the Sam68–PARP1–NFκB axis and enhances cell sensitivity to DNA damage.

### 3.4. ATM Modulates the RNA Binding Affinity of Sam68 upon DDR Induction

RBPs are increasingly recognized as key players in the DDR, and examples of coordinated regulation of DNA repair and RNA processing have been reported [6,7]. The RNA binding affinity of Sam68 and its activity in splicing and 3′-end processing are modulated by phosphorylation [39,54,55,56]. Thus, we asked whether post-translational modification by ATM affects the RNA binding affinity of Sam68. To address this question, HEK293T cells were transiently transfected with GFP-Sam68 in the presence or absence of Flag-ATM (Figure 5A,B; WCL panels). RNA pulldown assays showed that Sam68 specifically interacts with synthetic poly-A RNA and that activation of the DDR pathway, by either overexpression of Flag-ATM or treatment with NCS, increased its binding affinity (Figure 5A; polyA-Seph panels). Concomitant overexpression of Flag-ATM and treatment with NCS further increased the affinity of GFP-Sam68 for synthetic Poly-A RNA (Figure 5A; polyA-Seph panels). Moreover, the effect of ATM was inhibited by treatment with KU55933 (Figure 5B; polyA-Seph panels). To further test whether ATM-dependent phosphorylation was involved in this effect, we assessed the ability of Sam68 mutant isoforms to interact with poly-A synthetic RNA in the presence or absence of Flag-ATM (Figure 5C; WCL panels). Although all GFP-Sam68 mutants were able to bind RNA like the wild-type protein, mutation of T61 abolished the increase in binding observed in cells co-transfected with Flag-ATM, whereas the GFP-Sam68_S388A/S390A_ double mutant behaved similarly to the wild-type protein (Figure 5C; polyA-Seph panels). These results suggest that, in addition to promoting the ability of Sam68 to function in the DDR pathway, ATM-dependent phosphorylation of T61 increases the affinity of Sam68 for RNA, possibly impacting onits RNA processing activity.

### 3.5. ATM Promotes the RNA Processing Activity of Sam68

Sam68 regulates several aspects of RNA processing, including alternative splicing [39,46,55,56] and polyadenylation [29,57]. Since the splicing activity of Sam68 and other RBPs can be modulated by post-translational modifications [21,26,27,50,51] and ATM-mediated phosphorylation increases the affinity of Sam68 for RNA, we then sought to investigate whether it can also influence its RNA processing activity. To this end, we employed the CD44 exon v5 minigene as a model system to assess its splicing activity (Figure 6A). In this model gene, Sam68 was previously shown to promote inclusion of the exon v5 [46,56]. As expected, co-transfection with a suboptimal dose of Sam68 increased the inclusion of exon v5 in this splicing assay (Figure 6B). Interestingly, a small effect on exon v5 inclusion was also observed in HEK293T cells transfected with Flag-ATM. Furthermore, co-transfection of ATM with Sam68 further induced exon v5 inclusion (percentspliced in: PSI) (Figure 6B), indicating that ATM strengthens the splicing activity of Sam68. A similar effect was also observed using the Ppp3cc minigene (Appendix A), where Sam68 represses usage of an internal polyadenylation site in exon 14 and promotes selection of the last exon 16 (lower band in the gel) [57].

To test whether ATM-dependent phosphorylation of Sam68 was involved in the induction of its splicing activity, we performed splicing assays with the CD44 minigene and GFP-Sam68 mutants. As previously observed, co-transfection with ATM enhanced the effect of GFP-Sam68_wt_ on exon v5 inclusion (PSI 72± 1 vs. 60 ± 1), without affecting expression of GFP-Sam68 (Figure 6C). While a similar effect was also observed with the Sam68_S388A/S390A_ mutant (PSI 69 ± 2 vs. 61 ± 1), the T61A mutation completely abolished the effect of ATM without impairing the ability of Sam68 to function in splicing (PSI 67 ± 4 vs. 67 ± 3; Figure 6C). Collectively, these results indicate that phosphorylation of Sam68 at T61 by ATM increases its affinity for RNA and its splicing activity.

### 3.6. ATM Modulates Sam68 Function upon DDR Induction in Prostate Cancer Cells

Next, we asked whether the functional interaction between ATM and Sam68 could exert an impact on the post-transcriptional regulation of gene expression induced by DNA damage [7]. To test this hypothesis, we selected two genes (*PRKACB* and *SETMAR*) undergoing Sam68-mediated alternative polyadenylation regulation, which we hadrecently identified through RNA sequencing analysis of Sam68-depleted LNCaP cells (GSE198872). In both genes, the bioinformatics analysis predicted that Sam68 represses usage of an intronic polyadenylation signal (PAS), thus allowing expression of the full-length transcript that utilizes a PAS in the last coding exon (Figure 7A,C). Accordingly, quantitative real time PCR (qPCR) analysis showed that depletion of Sam68 increases selection of the internal PAS with respect to the distal PAS (Figure 7A,C). In line with an increased RNA processing activity of Sam68 upon genotoxic stress, NCS treatment caused the opposite trend, with increased selection of the distal PAS and production of the full-length transcript in both genes (Figure 7B,D). ATM depletion exerted an effect similar to Sam68 depletion on the regulation of these two alternative polyadenylation events. Moreover, ATM depletion completely suppressed the effect elicited by NCS (Figure 7B,D). These findings unveil the existence of a functional interplay between ATM and Sam68 in PC cells, which promotes both activation of the DDR pathway and modulation of RNA processing events in response to genotoxic stress.

## 4. Discussion

Mounting evidence suggests that ATM connects the DDR and RNA signaling pathways in response to endogenous and exogenous cues [58]. For instance, a bidirectional coupling between ATM signaling and the splicing machinery was observed in response to transcription-blocking DNA lesions and R-loop formation [14,58]. Moreover, several RBPs have been identified as putative substrates of ATM activity in response to DNA damage [59], thus supporting a key role for this kinase in the regulation of the RNA metabolism homeostasis upon genotoxic stress [58]. Our work now identifies a new molecular mechanism through which ATM contributes to the interplay between the DDR pathway and RNA-processing regulation. We found that activation of ATM, upon its overexpression, endogenous cues and/or DDR induction, promotes the interaction of the kinase with Sam68. This interaction leads to phosphorylation of Sam68 at canonical SQ/TQ consensus motifs located on the regulatory regions flanking its RNA binding domain. Sam68 was previously identified as a putative ATM substrate in large-scale studies aimed at identifying proteins containing phosphorylated SQ/TQ residues by immunoprecipitation and mass spectrometric analysis [48,59,60]. Nevertheless, the actual ATM-dependent phosphorylation of Sam68 and its functional consequences were not investigated. Herein, we identified T61, S388, and S390 as the Sam68 residues phosphorylated by ATM. Importantly, all three motifs identified in our study are evolutionarily conserved in mouse and rat, implying their functional relevance. Sam68 phosphorylation on these residues is induced by DNA damage upon treatment with the radiomimetic drug NCS and completely suppressed by the ATM-specific inhibitor KU55933. Moreover, our studies indicate that ATM-dependent phosphorylation of T61 promotes the function of Sam68 in the DDR pathway and enhances its RNA processing activity, whereas phosphorylation of S388 and S390 downstream of the RNA binding domain appears to exert milder effects. Thus, our work validates Sam68 as an ATM kinase substrate and uncovers a bidirectional interplay between ATM and Sam68, which couples the DDR pathway to modulation of RNA metabolism in response to genotoxic stress.

Sam68 was shown to support survival of PC cells exposed to DNA-damaging agents, such as etoposide or cisplatin [45]. However, the mechanism involved in this protective function was not elucidated. Subsequent studies in other cellular systems revealed a crucial role for Sam68 in promoting PARP1 recruitment to DNA lesions and NF-κB activation in response to genotoxic stress [42,43,44,53]. These studies indicated that the N-terminus of Sam68 (aa 1-101) is crucial for the interaction with PARP1 and that this interaction potentiated DDR induction. We now report that phosphorylation of T61 in Sam68 by ATM is required for strong activation of the DDR pathway in PC cells. In support of this notion, the Sam68_T61A_ mutant is impaired in its ability to interact with PARP1 and to activate NF-κB upon DDR induction, leading to increased sensitivity of PC cells to genotoxic stress. Although other RBPs were already shown to play critical functions in DDR signaling [61,62,63], Sam68 is the only one that is reported to act upstream of PARP1 activation [44]. Post-translational modifications and protein–protein interactions have been proposed to fine-tune PARP1 activity in the DDR pathway [64,65]. Likewise, Sam68 functions are extensively modulated by post-translational modifications [27]. However, the mechanism underlying Sam68-mediated activation of PARP1 at DNA damage sites was largely unknown. Our results fill this gap by providing evidence that ATM-dependent phosphorylation of Sam68 is required for its interaction with PARP1 and DDR induction upon DNA damage.

The splicing activity of Sam68 is enhanced by S/T phosphorylation on several residues located in the regulatory regions that flank its RNA binding motif [39,56]. Herein, we provide evidence that ATM-dependent phosphorylation of T61 increases the affinity of Sam68 for RNA and promotes its activity in alternative splicing and polyadenylation. More importantly, regulation of the RNA processing activity of Sam68 by ATM occurs in a physiological context, as activation of ATM upon DDR induction promoted alternative polyadenylation of two Sam68 targets in PC cells. Interestingly, knockdown of ATM in LNCaP cells exerted the same effect of Sam68 knockdown on these alternative polyadenylation events, indicating that basal ATM activity supports Sam68 function even in the absence of an exogenous stressor. This observation is in line with the detectable interaction between Sam68 and ATM in LNCaP under basal conditions, which may reflect low levels of ATM activation in response to endogenous DNA damage.

Several cancer cells exhibit persistent DDR activation [66]. On the other hand, upregulation of Sam68 expression is also frequent in human cancers [26,27], including PC [35,45]. Our study now couples these two events in a single scenario. We found that ATM interacts with and phosphorylates Sam68 in response to DNA damage, and that this functional interaction enhances activation of the DDR pathway. On the other hand, ATM-mediated phosphorylation of Sam68 increases its RNA processing activity, thus modulating expression of transcript variants that are targets of this RBP. It is tempting to speculate that this interplay between ATM and Sam68 may allow a homeostatic control of genome stability, by reinforcing the DDR pathway on one hand while maintaining the expression of proteins involved in the DNA repair on the other. For instance, we show that DDR induction in PC cells promotes productive splicing and polyadenylation of SETMAR, an enzyme that is crucial for repair and resolution of stalled DNA replication forks [67]. Sam68 prevents usage of a premature intronic polyadenylation site in the SETMAR pre-mRNA and ensures expression of the full-length transcript. Knockdown of either Sam68 or ATM unleashes this cryptic polyadenylation site and significantly reduces the expression of the full-length variant. The interplay between Sam68 and ATM may support proper expression of SETMAR, thus reducing the possibility of generating DNA lesions during the S phase of the cell cycle. PC cells, and possibly other cancer cells in which Sam68 is upregulated, may also exploit this physiological interplay to withstand DNA lesions elicited by chemotherapeutic drugs.

## 5. Conclusions

Our study demonstrates for the first time a direct molecular connection between the ATM kinase and the multifunctional RBP Sam68, which is likely implicated in the response of PC cells to genotoxic stress. These findings could pave the wayfor the development of tools that disrupt the ATM–Sam68 axis and sensitize cells to DNA damaging agents currently used in chemotherapy.

## Figures and Tables

**Figure 1 cancers-14-03847-f001:**
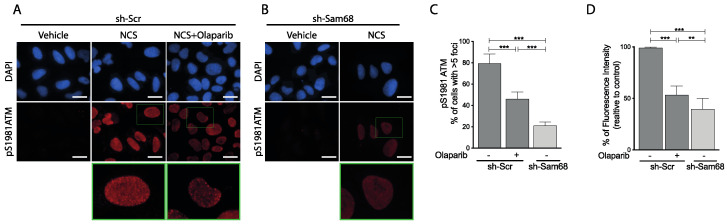
Sam68 is required for DDR activation in PC cells. (**A**,**B**) Representative immunofluorescence images of HEK293T cells infected with control (sh-Scr) lentivirus (**A**) or expressing RNA small interference for Sam68 (sh-Sam68) (**B**) treated, or not (vehicle), with NCS (30′, 500 ng/mL). Olaparib (10 µM, Selleck) was added 1 h before NCS treatment. After treatments, cells were fixed and stained with pS1981ATM antibody. (**C**,**D**) Bar graphs show the quantification of cells with more than five nuclear foci (**C**) and the fluorescence intensity relative to control (**D**). Bars, 10 μm, (n = 3; mean ± s.d.; ** *p* < 0.01, *** *p* < 0.001, *t* test).

**Figure 2 cancers-14-03847-f002:**
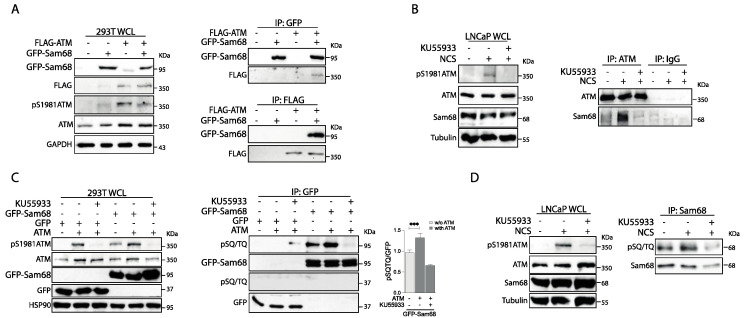
Sam68 interacts with ATM kinase, and it is phosphorylated on SQ/TQ motif upon DDR induction. (**A**) HEK293T cells were transiently transfected with a combination of different constructs that allow the overexpression of the fusion proteins GFP-Sam68 (pCDNA GFP-Sam68) and ATM-Flag (pCDNA FLAG-ATM) as indicated. Representative western blot of total protein extracts (whole-cell lysate, WCL) (293T WCL) and protein extracts subjected to co-immunoprecipitation assay. Co-immunoprecipitations were performed using anti-GFP (IP: GFP) or anti-FLAG (IP: FLAG) antibodies and immunoblotted with the indicated antibodies. (**B**) LNCaP cells were treated, or not, with KU55933 10 μM for 2 h and/or neocarzinostatin (NCS) 500 ng/mL for 1 h. Representative western blot of total protein extracts and protein extracts (LNCaP WCL) subjected to co-immunoprecipitation assay using anti-ATM (IP: ATM) antibody and IgG (IgG) as negative control. (**C**) HEK293T cells were transiently transfected as in (**A**) and treated, or not, KU55933 10 μM for 2 h. Representative western blot of WCL (293T WCL) and protein extracts subjected to immunoprecipitation assay using anti-GFP (IP: GFP) and immunoblotted with the indicated antibodies. Bar graph shows the densitometric analysis of pSQ/TQ signals with respect to GFP (western blot assays) in IP experiments (n = 3; mean ± s.d., *** *p* < 0.001, Student’s *t* test). (**D**) PC LNCaP cells were treated, or not, with KU55933 10 μM for 2 h and/or neocarzinostatin (NCS) 500 ng/mL for 1 h. Representative western blot of total protein extracts and proteins co-immunoprecipitated with anti-Sam68 antibody (IP: Sam68).

**Figure 3 cancers-14-03847-f003:**
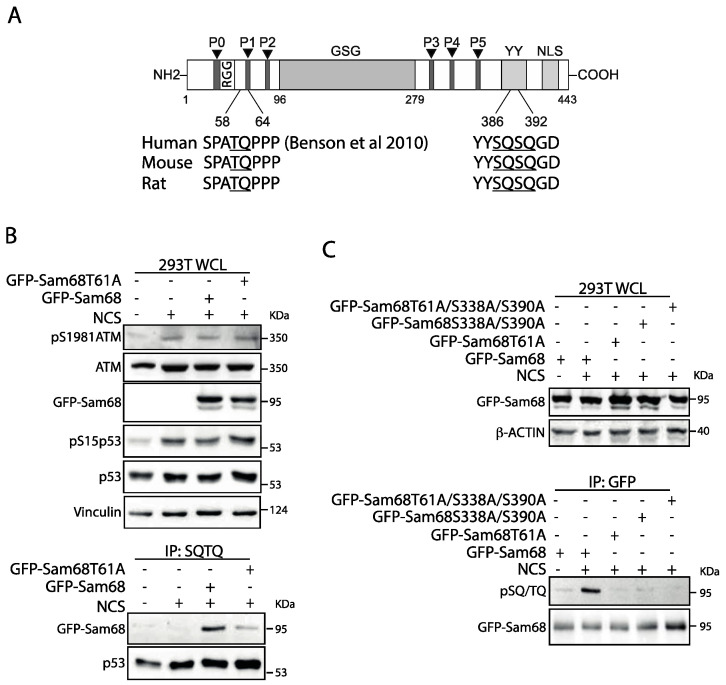
Preferential phosphorylation of T61 on Sam68 by ATM kinase activation upon DDR induction. (**A**) Schematic representation of putative serine and threonine residues on Sam68 phosphorylated by ATM kinase. (**B**) HEK293T cells were transiently transfected with fusion protein with either wild-type, GFP-Sam68_WT_, or mutants lacking threonine 61 phosphorylation sites, GFP-Sam68_T61A_, as indicated, and treated or not with NCS (500 ng/mL) for 1 h. Representative western blot of total protein extracts (whole-cell lysate, WCL) and protein extracts subjected to immunoprecipitation assay using anti-SQ/TQ antibody (IP: SQTQ) and immunoblotted with the indicated antibodies. Vinculin was evaluated as loading control of total protein extracts. (**C**) HEK293T were transiently transfected with either wild-type GFP-Sam68_WT_ or mutants lacking TQ/SQ phosphorylation sites, as shown in the schematic representation in (**A**), and treated, or not, with NCS (500 ng/mL) for 1 h. Representative western blot of total protein extracts (whole-cell lysate, WCL) and protein extracts subjected to immunoprecipitation assay using anti-GFP (IP: GFP) antibody and immunoblotted with the indicated antibodies. β-actin was evaluated as loading control of total protein extracts.

**Figure 4 cancers-14-03847-f004:**
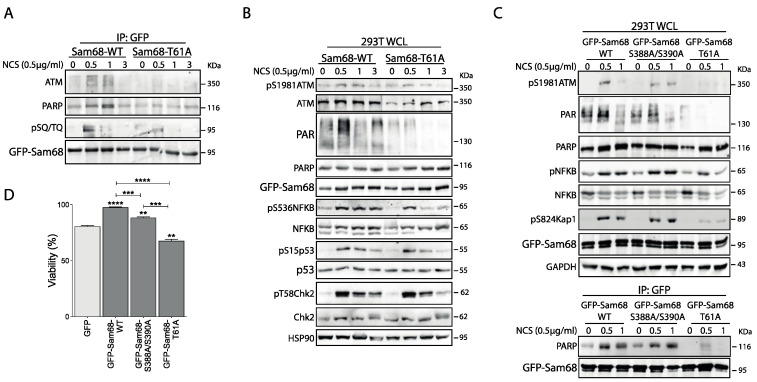
Sam68 phosphorylation on T61 is necessary for NF-κB activation and Sam68-PARP interaction upon DDR induction. (**A**,**B**) HEK293T cells were transiently transfected with fusion protein with either wild-type, GFP-Sam68_WT_, or mutant lacking threonin 61 phosphorylation site, GFP-Sam68_T61A_, and treated, or not, with NCS (500 ng/mL) and collected at the indicated time points (hours). (**A**) Representative western blot of protein extracts subjected to immunoprecipitation assay using anti-GFP antibody (IP: GFP) and immunoblotted with the indicated antibodies. (**B**) Representative western blot of total protein extracts (whole-cell lysate, WCL) and immunoblotted with the indicated antibodies. HSP90 was evaluated as loading control of total protein extracts. (**C**) Representative western blot of co-immunoprecipitation assay using anti-GFP (IP: GFP) antibody performed using WCL from HEK293T cells transfected with wild-type, GFP-Sam68_WT_, and GFP-Sam68_T61A_ and GFP-Sam68_S388A/S390A_ mutants, and treated, or not, with NCS (500 ng/mL) for the indicated time points (hours). (**D**) Bar graph represents viability assay performed in HEK293T cells transiently transfected, or not (GFP), with the indicated GFP-Sam68 plasmids and treated with NCS (500 ng/mL) (mean ± s.d., n = 3). After 48 h, cell viability was assessed by the Cell Titer Glo Luminescent Assay and reported as the percentage of cell viability with respected to untreated cells (100% viability). Statistical significance (** *p* < 0.01, *** *p* < 0.001, **** *p* < 0.0001; Student’s *t*-test) was evaluated with respect to GFP expressing cells treated with NCS (light grey bar). Brackets indicate additional statistical analysis of the indicated samples.

**Figure 5 cancers-14-03847-f005:**
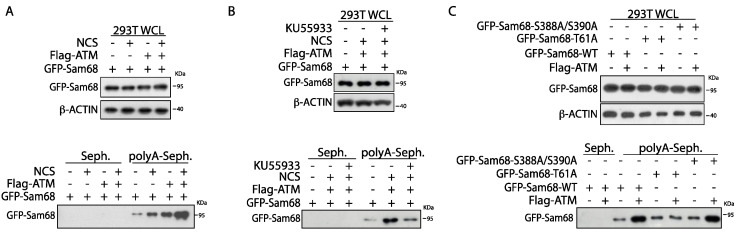
ATMmodulates Sam68 RNA binding affinity upon DDR induction. (**A**–**C**) Western blot analysis of pull-down assays performed using synthetic homopolymeric RNA. HEK293T cells were transiently transfected with indicated GFP-Sam68 plasmids and/or in addition to FLAG-ATM (**A**–**C**). After 24 h, cells have been treated, or not, with neocarzinostatin (NCS, 500 ng/mL) for 1 h (**A**) in presence, or not, of KU55933 (1 h) (**B**). Cell extracts (WCL) were incubated with either polyA-Sepharose (polyA) and streptavidine–Sepharose (Seph.) beads. Streptavidine–Sepharose beads have been used as negative control.

**Figure 6 cancers-14-03847-f006:**
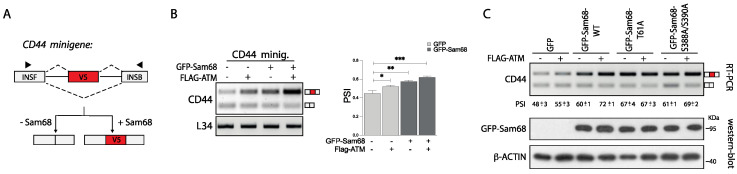
ATM-dependent phosphorylation modulates Sam68 splicing activity. (**A**) Schematic representation of CD44 minigene construct: v5 is the exon included in presence of Sam68 and black arrows indicate primers used for the RT-PCR analysis. (**B**,**C**) Representative PCR agarose gel of splicing assays of CD44 minigene performed in HEK293T cells transiently transfected with the indicated GFP-Sam68 plasmids and/or in addition to FLAG-ATM plasmid in presence of CD44 minigene (**B**,**C**). The percent spliced in (PSI) is reported in B and C (* *p* < 0.05, ** *p* < 0.01, *** *p* < 0.001, *t* test). L34 hasbeen used as a loading control (**B**). Western blot analysis to evaluate GFP-Sam68 proteins expression is also shown. β-actin was used as loading control (**C**).

**Figure 7 cancers-14-03847-f007:**
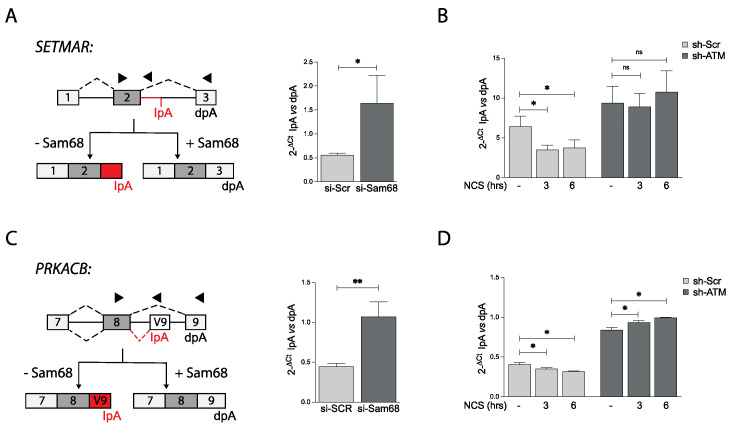
Sam68 APA activity is modulated upon DDR induction in an ATM-dependent way. (**A**–**D**) Bar graphs showing qPCR analyses of polyadenylation site (pA) usage evaluated in two representative genes (*SETMAR* and *PRKACB*) undergoing regulation in cells knocked down for Sam68 (si-Sam68) (**A**,**C**) or ATM (sh-ATM) and treated, or not, with neocarzinostatin (NCS, 500 ng/mL) for 1 and 6 h (**B**,**D**). Internal pA (IPA) usage was reported as 2^−ΔCt^ with respect to distal pA (d-pA) (* *p* < 0.05, ** *p* < 0.01, n.s. not significant, *t* test) (**A**–**D**). A schematic representation of SETMAR and PRKACB alternative polyadenylation events is also shown (**A**,**C**).

## Data Availability

The data presented in this study are available in this Appendix A.

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
