# Peer review of "DNA Damage Regulates the Functions of the RNA Binding Protein Sam68 through ATM-Dependent Phosphorylation"

_cancers, 2022, doi:10.3390/cancers14163847_

Round 1

Reviewer 1 Report

DNA and RNA metabolism, protein-protein interactions, and protein PTMs are usually altered in cancer cells, compared with normal cells. Emerging studies showed DNA damage signaling pathways modulate cellular autonomous metabolisms, including RNA metabolism. Nevertheless, the molecular mechanisms underlying remains unknown. In this context, the authors described that ATM phosphorylates Sam68 in response to DNA damage not only for DNA damage repair, but also for RNA processing. In prostate cancer, ATM mediated Sam68 phosphorylation determines its interaction with PARP1 and RNA, and also NFkB activation and targeted genes polyadenylation. However, the data should be strongly improved to support the arguments for publication.

Major:

  • Sam68 is required for DDR activation, particularly ATM phosphorylation. Meanwhile, Sam68 physically interact with phosphorylated ATM. Notably, ATM is kinase phosphorylating Sam68. This positive feedback look is interesting. The authors should better describe the biological relevance. For example, if Sam68 is required for ATM recruitment to DNA damage sites and vice versa.
  • In Fig 1B, it seems that KU55933 treatment reduces the ATM protein level. The authors should carefully check this abnormal phenotype.
  • In Fig 1C, the authors should note that the IP-western blot results are more striking than the quantitative results.
  • The authors revealed three S/T-Q motifs (T61, S388, S390) of Sam68 are important for ATM interaction. The authors can also construct the phosphor mimic mutation to check if they can enhance the ATM interaction, and also whether this interaction can be omitted by ATM inhibitor (KU55933). And it is also important to know if Sam68T61A compromises PARP1 recruitment to DNA damage sites.
  • It is confusion that which is responsible for Sam68 recruitment to DNA lesions, ATM or PARP? The authors should either deplete (or inhibit) PARP to examine this complex relationship between these three main players.
  • It is skeptical how Sam68T61A regulates ATM phosphorylation. In Figure 2, it seems Sam68T61A doesn’t alter AMT phosphorylation level. However, in Figure 3, Sam68T61A impairs ATM phosphorylation. It is also worth noting that Sam68S388A/390A stimulates ATM phosphorylation (Fig 3C). The authors should carefully confirm this important argument. Besides, the canonical ATM downstream protein should be looked at, such as KAP1 phosphorylation to confirm pATM function.  
  • In Fig 4A, the authors showed Flag-ATM can indeed increase Sam68 binding with RNA. The authors also need to check if this is dependent on ATM kinase activity. For example, the authors can treat the cells with KU55933 to check if the binding is reduced.
  • In Fig 4B (right panels), in terms of RNA pull down assay, the authors should include all samples in one gel for better comparison of RNA binding capability for different mutants.
  • The authors should carefully check the conclusion of “ATM-dependent phosphorylation modulates Sam68 splicing activity”, for several reasons. 1st, in Fig 5B, Sam68 indeed promote CD44 minigene splicing. However, the contribution of ATM is moderate. 2nd, in Fig 5C, the moderate (even minor) change from Lane 2-4 is not conclusive.     

Minor:

  • Introduction session (Paragraph 2): The authors described “DNA damage influences RNA transcription, processing and stability [8-10]” but the references are mainly focused on processing (splicing). The authors may also include those references for DNA damage influence RNA transcriptions and stability. The authors also need to confirm the proper citations throughout the manuscript.
  • “On the other hand, aberrant Sam68 expression has been correlated with malignant features and/or poor prognosis in a variety of human cancers [24,32,33] [34-37]” the references should be re-formatted.
  • In Paragraph 1, Page 11, NF-kB is not shown properly. 

Author Response

We have now addressed the issues raised by Reviewer 1. See attached file.

Reviewer 2 Report

The current study aims to look at the role of RNA binding protein Sam68 and its modulation by ATM kinase. The manuscript is well written and most of the comments are to help improve the flow of the manuscript that will help the readers better understand the experiments.

Figure 1C: What statistical test was used to analyze these graphs?

Figure 1: It is phosphorylated even in the absence of DDR. Modify the statement to say the phosphorylation is increased? Also, could the authors provide quantification for this increase as it's difficult to interpret this visually?

Figure 3A: Could the authors comment on why the interaction between PARP1 and SAM68 was reduced at 3hrs?

Figure 3B: The 0.5hrs time point shows pNFKb levels are higher compared to the other time points. Could the authors please shed some light on this?

Figure 3D: Could the authors comment on the different pair-wise comparisons done in and the rationale for the listed pair-wise comparisons?

Figure 3: What is the time unit?

Figure 4B: the authors indicate that the T61 mutation abolishes the positive effect of Flag-ATM. However, the difference in the band intensities of all 3 panels in figure 4B appears similar. Could the authors provide an explanation or clarification for the same?

Figure 5C: Again all the band intensities look very similar so could the authors please provide an explanation for this observation?

Could the authors please include a section in the materials and methods of statistical analysis used in the paper?

Also, not all panels presented in the manuscript are explained in the results section, especially the western blot data. It would be beneficial to the readers if the authors included a small description where necessary of all the panels of figures.

Author Response

We have now addressed the issues raised by Reviewer 2. See attached file.

Reviewer 3 Report

In this report the authors attempt to show that ATM phosphorylates Sam68 in response to DNA damage and this phosphorylation modulates Sam interaction with RNA and its involvement in splicing. Although, some figure panels appear to show aspects of this statement, there are major issues with this analysis. Sometimes the interpretations are weak or flat out wrong. I have debated whether to recommend rejection or major revisions. Please see my comments below. I leave it to the editor to make this decision.

Major comments.

1.       Supplementary Fig S1. The result for LNCaP cells is interesting but also puzzling. First, it appears that SAM68 silencing was not very good which may explain why you still see ATM phosphorylation. This is interesting because it directly connects SAM68 with ATM phosphorylation (e.g., the more you downregulate SAM expression the less ATM phosphorylation is seen) and should be discussed. Second, why is ATM phosphorylation disappearing after 6hrs of treatment? Is this due to checkpoint attenuation? Or is this due to the drug efficiency (e.g. drug half-life, degradation, etc).

2.       Figure 1A. The authors claim that over-expression of SAM-GFP results in activation of ATM but this reviewer is not convinced. ATM is phosphorylated even in the absence of GFP-SAM (see line 3). In fact, more that band is stronger than the one in lane 4. How do the authors make this conclusion? To this reviewer it seems that over-expression of GFP-SAM has no effect on ATM phosphorylation. In fact, just probing for ATM detects the phosphorylation moiety. In fact, this is not unexpected because trace amounts of phosphorylated ATM would occur in cycling cells especially in the HEK293T cell line.  I think the authors should revise their interpretation.

3.       Figure 1C right panel. Is the phosphorylation of SAM by ATM occurring in the absence of the NCS mimetic? This is an important finding which indicates that the ATM dependent phosphorylation is not DNA damage induced. This is clear from panel 1D (right). And this is not in over-expressed SAM-GFP, therefore directly contradicting the statements made about panel 1A. This actually proves my point which is that ATM activation occurs due to some other form of DNA damage, perhaps endogenous damage occurring in this cell line and has nothing to do with SAM over-expression. Thus, this sentence is incorrect: “These results identify Sam68 as a new ATM-interacting protein which is phosphorylated upon activation of the kinase by DDR induction.” Phosphorylation of SAM occurs whether or not you have DDR induction. The ATM inhibitor definitely indicates that is ATM dependent but I am not sure that it is DDR induced. In fact, panel 1D shows that ATM does not even have to be activated (e.g. S1981 phosphorylation) in order to phosphorylate SAM.

4.       Figure 2C. The authors argue that “the SQ motifs downstream of the GSG are also substrates of ATM activity..” but this does not appear to be so. If you look at Fig.2C pSQ blot mutation of just T61A is sufficient to kill all protein phosphorylation. Then again co-mutation of S338 and S390 also kills protein phosphorylation. Finally same result is seen when all three residues are mutated. This is quite puzzling. It shows that no matter what mutation is made, phosphorylation is abrogated. This is not due to the fact that protein expression is affected in the mutant samples because the WCL blot above shows that GFP Sam is being made. This reviewer is not convinced by the interpretation of these results (e.g. “preferential phosphorylation of T61…).

5.       Fig. 3A. Here pSQ phosphorylation seems to happen even in the T61A mutant. Unfortunately, the blot is not very clear but I definitely see a band at 0.5hrs. From what I see here, the interaction with PARP is stronger when SAM is not phosphorylated (compare SAM68 WT 0.5 with 1). Also, is SAM marked with GFP here? Is this the same construct as in figure 2? Please label correctly. Also, the authors say that “ATM activation is correlated with SAM-PARP interaction” but is the blot showing phosphorylated ATM or just total ATM? Again, this goes back to my previous point that ATM activation is not required for SAM phosphorylation.

6.       Fig 3B. The NF-KB story is also a bit odd. It is phosphorylated at 0.5NCS in Sam WT and SAM T61A. However, phosphorylation quickly decreases in the mutant but not in the WT. The conclusion here is that mutant SAM cannot sustain activated NFKB, or p53 and Chk2 because all three behave the same. I agree with the conclusion that a decrease in SAM-PARP interaction exists in T61A mutant. I would argue that the interpretation of 3B and 3C is that the reason NFKB, p53 and CHK2 phosphorylation cannot be sustained in SAM68T61A is because of the decreased interaction between SAM-PARP. The viability graph in 3D further shows that T61A is the key residue with the other residues making minor contributions.

7.       Fig 4B. “We observed that mutation of T61 abolished the positive effect of Flag-ATM on the affinity of GFP-Sam68 for synthetic RNA, whereas the GFP-Sam68S388A/S390A double mutant behaved similarly to the wild-type protein (Figure 4B).” Are the authors referring to a different figure? I see two bands in every blot. Perhaps the affinity is slightly decreased but definitely not abolished. Please point to the blots where you see “abolishment”. In fact, I would argue that there is almost no effect.

8.       Fig. 5B. “Furthermore, co-transfection of Sam68 and ATM strongly induced exon v5 inclusion (Figure 5B), indicating that up-regulation of ATM strengthens the splicing activity of Sam68.” First of all, I wouldn’t say “strongly”. The band is somewhat darker than in lane 3. Second, what do you mean by “strengthens the splicing activity”? In fact there is less splicing in the FLAG-ATM. I would argue that co-expression of SAM and ATM inhibits V5 splicing. Also, the opposite effect is seen in PPP3CC. You see no splicing of pA when co-expressing ATM and SAM. These two figures taken together with Figure 4B makes these results completely uninterpretable: We don’t know what the effect of co-expression is and we don’t know if Sam phosphorylation affects is RNA biding activity.

9.       Now Fig.6 does appear to show that Sam controls splicing of these genes but is not clear whether is due to its phosphorylation by ATM because this is not shown. So the only conclusion here is that certain RNA splicing is affected by Sam expression but we don’t know if it is due to phosphorylation or even RNA biding.

Minor comments

1.       First line of abstract “DNA damage response (DDR) response,…” Why use “response” twice?

2.       All figures should be enlarged but particularly figures 1, 3, and 5. I suggest that figures 1 and 3 should be spread out on an entire page. This journal does not have page limitations and there is no reason to crowd them all in a small format. The supplementary figures should also be significantly enlarged.

3.       I am confused by panel 1C. Is one of the GFP blots looking for GFP-SAM and the other for just GFP? I assume GFP vector was used as control and this appears to be so from the size of the band but better labeling should be used. Perhaps the top of the panels should be labeled GFP-vector or something like that. This should also be described more thoroughly in the lengend.

Author Response

We have now addressed the issues raised by Reviewer 3. See attached file.

Round 2

Reviewer 1 Report

Thanks the authors for addressing the major points in this time frame. 

Regarding to Figure 1, please quantify the signal intensity rather than foci number more than 5. 

Author Response

As requested by the Reviewer, we have now added an additional panel (Figure 1D) with quantification of the overall nuclear intensity of fluorescence. However, as accumulation of ATM in foci is a hallmark of DDR activation, we also prefer to maintain the quantification of cells with >5 foci/ nu lets (Figure 1C). The text has been changed accordingly.

Reviewer 2 Report

I thank the authors for their response. The authors have addressed the comments from this reviewer in a satisfactory manner.

Author Response

We wish to thank the Reviewer for her/his comments.

Reviewer 3 Report

The authors have made major revisions to this version or provided compelling arguments for their interpretation. This reviewer is satisfied.

The only suggestion I would make is to include in the text of the paper the language they used in response to my comment 8. Perhaps starting at "...splicing factors insert different actions..." and provide those references in the paper. The argument made for point 8 is compelling and I think that it should be in the paper because it will help readers understand the interpretation of these results.

Otherwise, this reviewer is satisfied.

Author Response

As suggested, we have now added the information regarding the different actions of splicing factors in splicing regulation, with relative references.